# An Enhanced Three-Dimensional Auxetic Lattice Structure with Improved Property

**DOI:** 10.3390/ma13041008

**Published:** 2020-02-24

**Authors:** Yingying Xue, Peixin Gao, Li Zhou, Fusheng Han

**Affiliations:** 1School of Electromechanical and Automotive Engineering, Yantai University, Yantai 264005, China; Xyy_0806@126.com (Y.X.); peigaox@ytu.edu.cn (P.G.); 2Key Laboratory of Materials Physics, Institute of Solid State Physics, Chinese Academy of Sciences, Hefei 230031, China

**Keywords:** enhanced 3D auxetic lattice structure, enhanced stiffness and strength, 3D re-entrant lattice structure, compressive property, negative Poisson’s ratio, 3D printing

## Abstract

In order to enhance the mechanical property of auxetic lattice structures, a new enhanced auxetic lattice structure was designed by embedding narrow struts into a three-dimensional (3D) re-entrant lattice structure. A series of enhanced lattice structures with varied parameters were fabricated by 3D printing combined with the molten metal infiltration technique. Based on the method, parameter studies were performed. The enhanced auxetic lattice structure was found to exhibit superior mechanical behaviors compared to the 3D re-entrant lattice structure. An interesting phenomenon showed that increasing the diameter of connecting struts led to less auxetic and non-auxetic structures. Moreover, the compressive property of the enhanced structure also exhibited obvious dependence on the base material and compression directions. The present study can provide useful information for the design, fabrication and application of new auxetic structures with enhanced properties.

## 1. Introduction

As a novel metamaterial, auxetic structures possess unconventional properties, i.e., they expand laterally under tension and shrink under compression, which are also described by negative Poisson’s ratio. This unique behavior offers enormous improvement in the mechanical behavior of the structures, such as excellent shear stiffness [1,2], enhancement in fracture toughness [3], indentation resistance [4], high energy dissipation [5] and outstanding acoustic absorption abilities [6,7,8]. Therefore, auxetic structures have a wide range of applications in aerospace [9], biomedical engineering [10,11], sensors and actuators [12] and functional structures [2]. 

A large-scaled auxetic structure was first attained in the form of 2D silicone rubber or aluminum honeycombs deforming by flexure in 1982 [13]. Studies on auxetic materials have gained interest as a result of the pioneering works of Lakes [3] and Evans and Caddock [14]. Nowadays, more and more auxetic structures are being designed, fabricated and characterized. Gibson et al. [13] studied the deformation behavior of honeycomb structure including elastic buckling, the plastic collapse behavior and the formation of plastic hinges of cell walls. The chiral structures which were subjected to uniaxial in-plane force were studied and the in-plane Poisson’s ratio and elastic modulus were gained experimentally and numerically [15]. Recently, a re-entrant auxetic structure was fabricated and the failure mode was obtained under compression. A deflection analytical model was also proposed to predict the structure’s yield strength and modulus [16]. Although a lot of research is being carried out on auxetic lattice structures, the existing auxetic lattice structures still exhibit limitations, i.e., low stiffness and strength due to high porosity and the bending or rotation deformation nature of the struts, which limit the application of auxetic lattice structures. Considering this, it is really important to find effective methods to improve the stiffness and strength of auxetic lattice structures. Regarding the enhancement of stiffness and strength, gradient honeycomb has broad application prospects in enhancing the in-plane stiffness and physical characteristics, and also has broad application prospects in sandwich structures [17,18,19]. Auxetic reinforced composites containing inclusions have also been verified to exhibit potential in enhancing structure characteristics [20]. Recently, some narrow struts have been added to the 2D re-entrant honeycomb to improve its stiffness and strength, and embedding the enhanced structure into a 2D re-entrant hexagon cell has actually been confirmed to be an effective method to improve its mechanical properties [21,22]. 3D auxetic lattice structures are as useful as 2D structures, and sometimes exhibit advantages that the latter do not have. In addition, the progress of manufacturing technology makes it possible to manufacture 3D structures with an arbitrary complex microstructure. Therefore, more and more three-dimensional structures have been fabricated and studied. 

In order to enhance the strength of a 3D re-entrant lattice structure, several narrow struts were added into the structure and the enhanced 3D auxetic structure was fabricated by 3D printing combined with the molten metal infiltration technique. Based on this method, parameter studies of the enhanced auxetic lattice structure were carried out to explore the relationship between structure parameters and its properties. The effect of the base material and compression directions on the properties of the enhanced structure were also studied. It is expected that the analysis in this article may provide a good way for enhancing the mechanical properties of lattice structures. 

## 2. Experimental

### 2.1. Structure Design

A 3D re-entrant unit cell was constructed by connecting a 2D unit cell with cylindrical struts (Figure 1a,b). Similarly, a novel 3D enhanced unit cell can be readily attained through the combination of a 2D enhanced unit cell (Figure 1c,d). Then, the 3D structure was produced through CATIA V5 R20 (Figure 1e). Parts f and g of Figure 1 show the three views of the unit cells of the 3D re-entrant lattice structure and the enhanced 3D structure, respectively. In this study, we examined the compressive characteristics of the enhanced lattice structure experimentally.

The enhanced auxetic structure parameters include the side strut length, *H*; the inner strut length, *L*; the connecting strut length, *L*_1_; the angle of inner strut with side strut, *θ*; the angle of connecting strut with the horizontal direction, *θ*_1_; the diameter of side and inner struts, *D*; and the connecting strut diameter, *d* (Figure 2). It is noteworthy that the value of *L*_1_ and *θ*_1_ depend on other parameters which cannot be changed independently. As shown in Figure 2, the lattice geometric parameters have the relationship as follows:(1)$$L_{1}=L\sqrt{{\left[\frac{H}{2L}-cos\theta \right]}^{2}+sin^{2}}\theta \;$$
(2)θ1=arctan[H2L−cosθsinθ] 

The relative density of the enhanced 3D auxetic structure and 3D re-entrant structure can be yielded by:(3)$$\begin{array}{l} \bar{\rho }=\frac{\rho_{0}}{\rho_{s}}\\ =\;\frac{4\pi {\left(\frac{D}{2}\right)}^{2}H^{\prime }+16\pi {\left(\frac{D}{2}\right)}^{2}L^{\prime }+24\pi {\left(\frac{d}{2}\right)}^{2}L_{1}}{2\left(H-Lcos\theta \right){\left(2Lsin\theta \right)}^{2}}\;=\;\frac{\;4\pi {\left(\frac{D}{2}\right)}^{2}\left[H+\frac{\left(1-cos\theta \right)}{sin\theta }D\right]+16\pi {\left(\frac{D}{2}\right)}^{2}\left(L-\frac{D}{sin\theta }\right)+24\pi {\left(\frac{d}{2}\right)}^{2}L_{1}}{2\left(H-Lcos\theta \right){\left(2Lsin\theta \right)}^{2}}\\ =\;\frac{\left[\alpha +4-\frac{(3+cos\theta }{sin\theta }\frac{)D}{L}\right]\pi D^{2}+6\pi d^{2}\frac{L_{1}}{L}}{8\left(\alpha -cos\theta \right){\left(Lsin\theta \right)}^{2}} \end{array}$$
and
(4)$$\begin{array}{ll} {\bar{\rho }}_{1}=\;\frac{\rho_{o}}{\rho_{s}}\;=\;\frac{4\pi {\left(\frac{D}{2}\right)}^{2}H^{\prime }+16\pi {\left(\frac{D}{2}\right)}^{2}L^{\prime }}{2\left(H-Lcos\theta \right){\left(2Lsin\theta \right)}^{2}} & =\;\frac{\;4\pi {\left(\frac{D}{2}\right)}^{2}\left[H+\frac{\left(1-cos\theta \right)}{sin\theta }D\right]+16\pi {\left(\frac{D}{2}\right)}^{2}\left(L-\frac{D}{sin\theta }\right)}{2\left(H-Lcos\theta \right){\left(2Lsin\theta \right)}^{2}}\\ & =\;\frac{\left[\alpha +4-\frac{(3+cos\theta }{sin\theta }\frac{)D}{L}\right]\pi D^{2}}{8\left(\alpha -cos\theta \right){\left(Lsin\theta \right)}^{2}} \end{array}$$
where *ρ*_0_ is the density of all struts in a unit cell, *ρ*_s_ is the apparent density of a unit cell and *α* is the designed length ratio of side strut to inner strut, i.e., *α* = *H*/*L*.

In order to reveal the compressive behavior and enhancement of the structures, several samples were designed with variable parameters, as shown in Table 1 and Table 2.

### 2.2. Fabrication of Lattice Samples

The preparation approach used in the present study is the same as that reported in ref. [23]. Firstly, the auxetic structures with varied parameters were prepared by 3D printing based on photosensitive resin (Figure 3a). Subsequently, the auxetic lattice pattern based on the photosensitive resin was put into a container and the plaster slurry with appropriate composition and viscosity was poured into the container to fill the cells of the auxetic structures. After the plaster slurry dried and hardened naturally, a series of treatments were adopted to burn away the photosensitive resin, producing a porous framework in a mold. Finally, molten aluminum was poured into the mold and made to penetrate into the interstice of the porous framework under external pressure. Then the composite was sprayed to obtain the 3D Al-based enhanced auxetic structure (Figure 3b), which has same framework with the original photosensitive resin auxetic lattice structure.

### 2.3. Mechanical Measurement

The compressive behavior of the samples was tested by the Instron 3369 measuring system with the compression rate of 2 mm/min. The strain range was from 0 to 60% or over to guarantee the observation of full compression deformation process. As shown in Figure 3b, there were five unit cells in the X and Y directions and three in the Z direction. For each structure, at least three samples were tested and the arithmetic mean value was used as the representative value for each sample group. The middle area displacement of the samples was tested by the Image J software to calculate the Poisson’s ratios of the structures. Then the Poisson’s ratio *υ_zy_* could be calculated by the following formula:(5)$$v_{zy}=-\frac{\varepsilon_{y}}{\varepsilon_{z}}$$
where $\varepsilon_{y}$ and $\varepsilon_{z}$ are lateral strain and vertical strain, respectively.

## 3. Results and Discussion

### 3.1. Parameter Studies of the Structures

#### 3.1.1. Effect of Strut Diameter d

Figure 4a shows the engineering stress–strain curves of samples with variable connecting strut diameters. The curves were similar to those of other lattice structures and porous metallic materials [24,25], i.e., they consisted of three regions including the initial elastic region, plateau region and densification region. The appearance of each region is related to the mechanisms of deformation and interactions among the struts as shown in Figure 5, which has been extensively discussed by researchers. As expected, both the elastic modulus and compressive strength increased with increasing connecting strut diameter. When the *θ* angle, strut diameter and length were the same, the enhanced lattice structures showed obviously higher compressive stress compared with re-entrant structures, although the relative density of enhanced structures increased, for example, in samples 3 and 8. It can be seen from Figure 4a that the stresses for sample 3 and sample 8 were 8.1 MPa and 2.1 MPa at the strain of 0.3, respectively. The relative density increased by less than one time and the stress increased by more than three times. This enhancing effect would mainly result from the resistance of connecting struts to the deformation of inner struts, and can be easily understood from the structure, as shown in Figure 2, in which the connecting struts form several triangles with inner and side struts. All the angles of these triangles were acute angles with the z-axis or compression direction. Obviously, the enhancing connecting struts markedly increased the strength of the whole lattice structure and the bending of inner struts became much more difficult, which, correspondingly, reduced the Poisson’s ratio of the structure. As shown in Figure 4b, with increasing connecting strut diameter, the Poison’s ratio of the enhanced 3D structure changed from negative to positive, which indicated that the connecting struts would reduce the auxetic behavior of the structure.

#### 3.1.2. Effect of θ Angle

Figure 6a and Figure 7 show the stress–strain curves of the structures with variable *θ* angles. The compressive strength and elastic modulus of the enhanced structure and 3D re-entrant structure decreased with increasing *θ* angle, while the plateau regions elongated if other parameters were kept constant. This should be attributed to the decreased relative density of the structures, as shown in Table 1 and Table 2. In addition, it is worth noting that slight drops in the stress-strain responses of the 3D re-entrant structure can be observed during the plateau regions (Figure 7). This is due to the instability of the re-entrant structures during the compressive process, which will cause local stress maximization, leading to failure. While all the stress–strain curves of the enhanced auxetic lattice structures were smooth in character and did not exhibit undulations, which is due to the stability of the structures and uniform stress distribution in enhanced lattice structures compared with that of the 3D re-entrant structures, which effectively suppressed the stress concentration. This showed that embedding narrow struts into each cell of 3D re-entrant lattice structures can also enhance the stability of the structures.

Due to the enhancement of the connecting struts, the absolute value of the Poisson’s ratios of the enhanced auxetic structures were smaller than that of the 3D re-entrant structures (Figure 6b). It is worth noticing that the absolute value of Poisson’s ratios of the enhanced lattice structures monotonically increased with angle *θ*, which was consistent with the re-entrant structures. The reason is that, when the angle *θ* increases, the inner struts of both the enhanced structure and 3D re-entrant structure endure higher axial force compared to struts with a lower angle, resulting in higher structural deformation and Poisson’s ratio. Besides, due to the overlapping of the side strut and the inner strut, the effective inner strut length for bending is shorter than the designed value. The larger the angle *θ*, the less the reduction of the struts and the higher the absolute value of Poisson’s ratio. These influences could be the reason for the decrease in the negative Poisson’s ratios of structures.

Figure 8a,b shows the experimental result of Poisson’s ratios of the enhanced auxetic structures and 3D re-entrant structures during the compression tests. The Poisson’s ratios of both the structures exhibited decreasing characteristics with the compressive deformation process. This is due to the fact that, from the strut arrangement shown in Figure 1, the vertical deformation should be easier than horizontal deformation because the vertical deformation is resisted by the bending of inner struts, while the horizontal deformation is predominantly impeded by the axial stress of side struts, resulting in smaller horizontal strain and larger vertical strain and decreasing the absolute value of Poisson’s ratios. Due to the continuous compression of the lattice structure, the compressive deformation becomes much more difficult because of the more compact of the structure; this is another reason for the reduction of the absolute value of Poisson’s ratios of the structures.

### 3.2. Effect of Compression Direction

The compressive properties of enhanced auxetic lattice structures with the same geometrical parameters and unit cells, but different compression directions, were studied. The stress compressed in the X/Z direction was higher than that in the Y direction, and the plateau region was longer in the former than in the latter (Figure 9a,b). It can be concluded that the compression direction significantly influenced the response of the enhanced lattice structures, indicating an anisotropic characteristic of the present auxetic lattice structures. This can be related to the deformation patterns along the different directions. When compressed along the Y-axis direction, due to the finite number of unit cells, some side struts are subject to the bending moment. As a result, the vertical struts will tend to yield first due to the edge effect, which lowers the stress. Moreover, the connecting struts move toward the horizontal direction, which makes the struts contact with each other easily. As a result, the densification initiates earlier.

### 3.3. Effect of Material

The stress–strain curves of sample 3 with the same structural parameters, but different base materials, were plotted to study the influence of base materials on the mechanical behaviors of the structure (Figure 10). It is worth noting that the base materials significantly influenced the compressive behavior of the enhanced lattice structure. Although the stress–strain curves exhibited a similar trend, the appreciable stress oscillation occurred in the plateau region of the 6063 Al alloy lattice structure, which indicated the unstable deformation characteristic related to the base material behavior. This is because the 6063 Al alloy does not show much plasticity before fracture and some of the struts will fracture first and then the stress will be distributed in other struts, resulting in the peaks in the plastic region. Due to the excellent ductility of pure Al (99.6%), the stress–strain curve of the Al-based lattice structure showed a smooth character. The compressive strength and elastic modulus of the 6063 Al alloy lattice structure were much higher than the Al lattice structure. It can be therefore summarized that the compressive mechanical property of lattice structures can also be improved through the base material.

### 3.4. Analysis of Enhanced Mechanical Properties

Typically, it is necessary to maximize the specific stiffness and strength, i.e., $\left(E/E_{s})/(\rho_{0}/\rho_{s}\right)$ and $\left(\sigma /\sigma_{s})/(\rho_{0}/\rho_{s}\right)$, of the cellular structural designs [26]. Here, the specific stiffness of the enhanced lattice structure and the 3D re-entrant lattice structure were $\;(E/E_{s})/\bar{\rho }$ and $(E_{1}/E_{s})/{\bar{\rho }}_{1},\;$ respectively, and the specific strength of the enhanced lattice structure and 3D re-entrant lattice structure were $(\sigma /\sigma_{s})/\bar{\rho }$ and $(\sigma_{1}/\sigma_{s})/{\bar{\rho }}_{1},$ respectively. The enhancement of the specific stiffness and strength of the enhanced lattice structures with respect to the 3D re-entrant lattice structures is given by:(6)(E/Es)/ρ¯(E1/Es)/ρ¯1 = Eρ¯1E1ρ¯
(7)(σ/σs)/ρ¯(σ1/σs)/ρ¯1 = σρ¯1σ1ρ

Figure 11a,b shows the enhancement of the specific stiffness and specific strength of the enhanced lattice structures with different connecting strut diameter d and angle θ. The enhancement increased with increasing connecting strut diameter d, while it decreased with increasing angle θ, and the specific stiffness and strength of the enhanced lattice structure were improved compared to the 3D re-entrant structure. The relative enhancement of the specific strength ranged from 3.7 to 4.5 and 3.5 to 4.3 compared to 3.4 to 4.2 and 3.2 to 4.0 for specific elastic modulus. Therefore, the enhanced lattice structure was more efficient to improve specific strength rather than stiffness. This is because the compressive strength is significantly influenced by the local plastic yielding of the struts and the narrow struts can effectively stiffen the structure and suppress the local deformation of the structure, so it is more efficient to improve the compressive strength of the enhanced structure; as a result, the specific strength exhibits much more enhancement. Therefore, it is expected that this should provide a good way for enhancing the mechanical properties for lattice structures.

In order to compare the behaviors of the enhanced lattice structures with other cellular materials built using various materials, the strength values were normalized by dividing the bulk material strength (Figure 12). The enhanced structures exhibited higher compressive strength than most of the other lattice structures and foam structure, indicating that the enhanced structures can be used for many potential applications.

## 4. Conclusions

An enhanced 3D auxetic lattice structure was proposed and fabricated through 3D printing combined with the molten metal infiltration technique in this study. The enhanced structure exhibited a three-region compressive mechanical characteristic, i.e., a linear elastic region, a plateau region, and a slightly increasing and densification region, similar to common cellular materials. The effect of the connecting struts’ diameter d and angle θ on the compressive mechanical behaviors of the enhanced auxetic structures was thoroughly investigated. It was shown that both of them exhibited significant effects on the mechanical property of the structures. A comparison between the newly proposed auxetic lattice structure and the 3D re-entrant lattice structure was also performed, which showed that the enhanced auxetic lattice structure had superior performance features.

## Figures and Tables

**Figure 1 materials-13-01008-f001:**
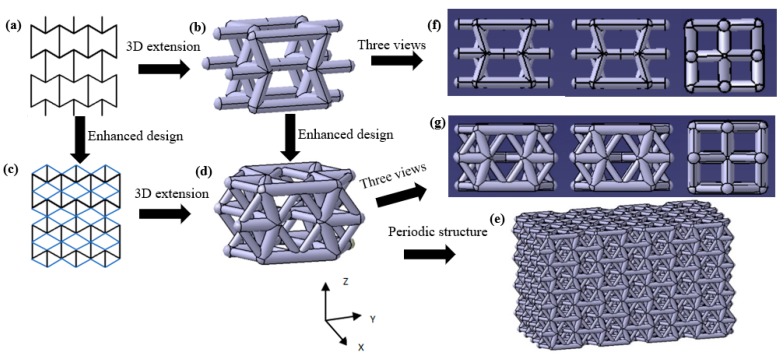
Design route of the enhanced auxetic lattice structure: (**a**) 2D unit cell, (**b**) 3D unit cell of re-entrant lattice structure, (**c**) 2D enhanced unit cell, (**d**) unit cell of the enhanced 3D structure, (**e**) enhanced 3D auxetic structure based on corresponding unit cell, (**f**) three views of the unit cell of the 3D re-entrant structure, (**g**) three views of the unit cell of the enhanced 3D structure.

**Figure 2 materials-13-01008-f002:**
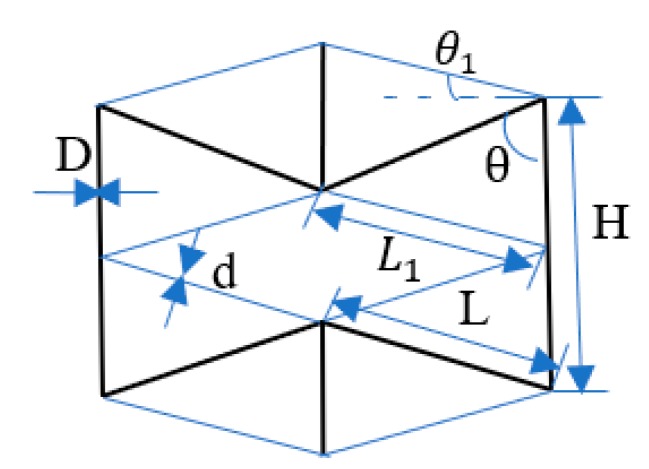
Design parameters of the enhanced auxetic lattice structure.

**Figure 3 materials-13-01008-f003:**
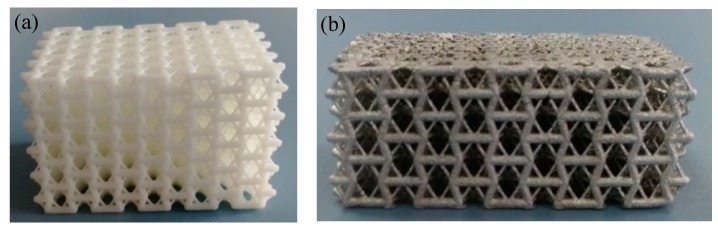
(**a**) 3D auxetic structure based on a photosensitive resin and (**b**) lattice sample based on aluminum.

**Figure 4 materials-13-01008-f004:**
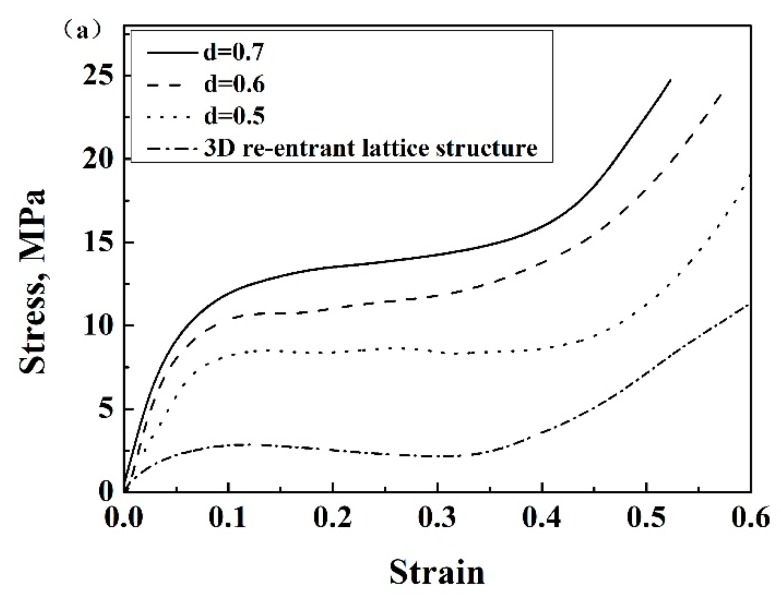
(**a**) Stress–strain curves of enhanced lattice structures with different connecting strut diameters d based on Al, (**b**) Poisson’s ratios of enhanced structures with different connecting strut diameters d based on Al.

**Figure 5 materials-13-01008-f005:**
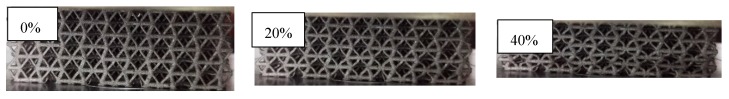
The deformation modes of the structure at different strain regions.

**Figure 6 materials-13-01008-f006:**
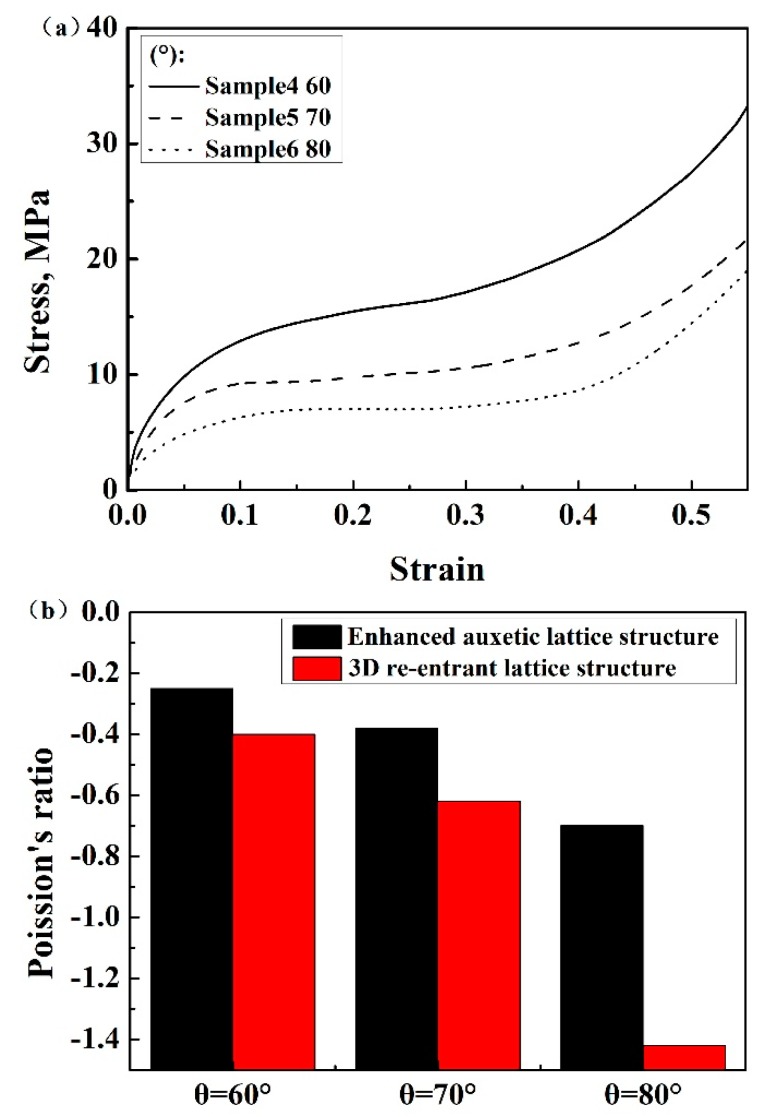
(**a**) Stress–strain curves of enhanced structures with different *θ* angles based on Al, (**b**) Poisson’s ratios of enhanced structures and 3D re-entrant lattice structures with different *θ* angles based on Al.

**Figure 7 materials-13-01008-f007:**
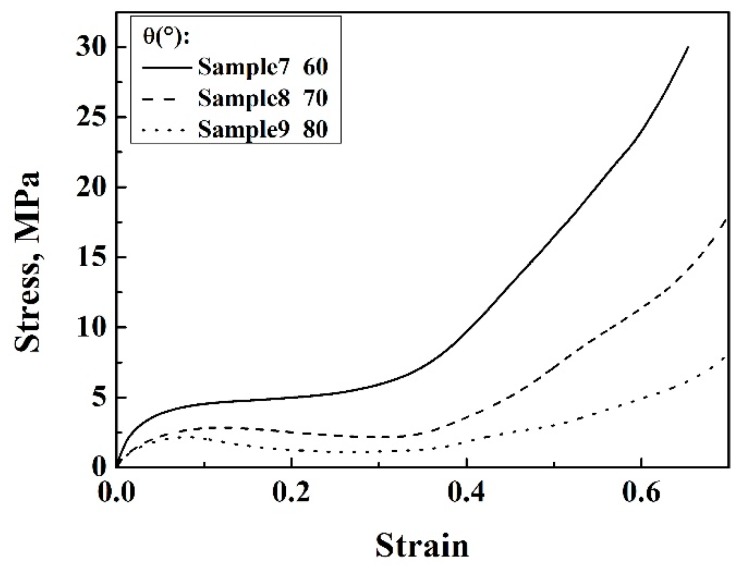
Stress–strain curves of 3D re-entrant lattice structures with different *θ* angles based on Al.

**Figure 8 materials-13-01008-f008:**
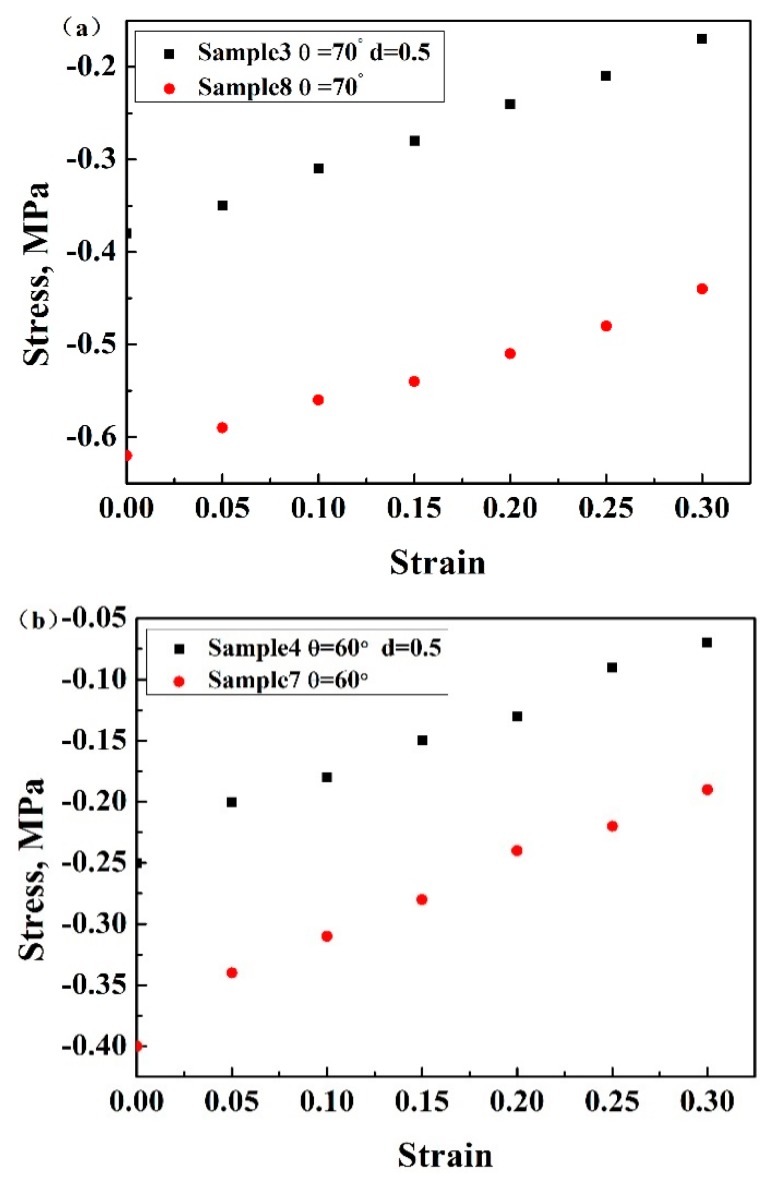
Poisson’s ratios of (**a**) sample 3 and sample 8, (**b**) sample 4 and sample 7 based on Al.

**Figure 9 materials-13-01008-f009:**
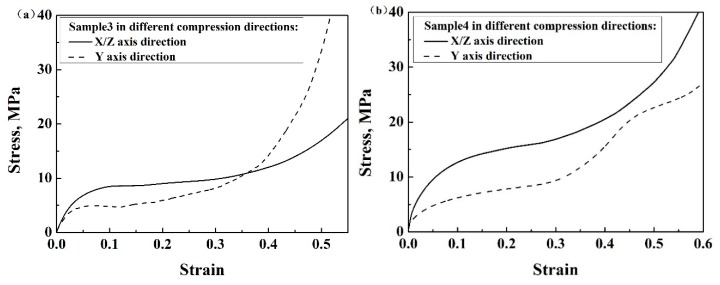
Stress–strain curves of (**a**) sample 3 and (**b**) sample 4 based on Al in different compression directions.

**Figure 10 materials-13-01008-f010:**
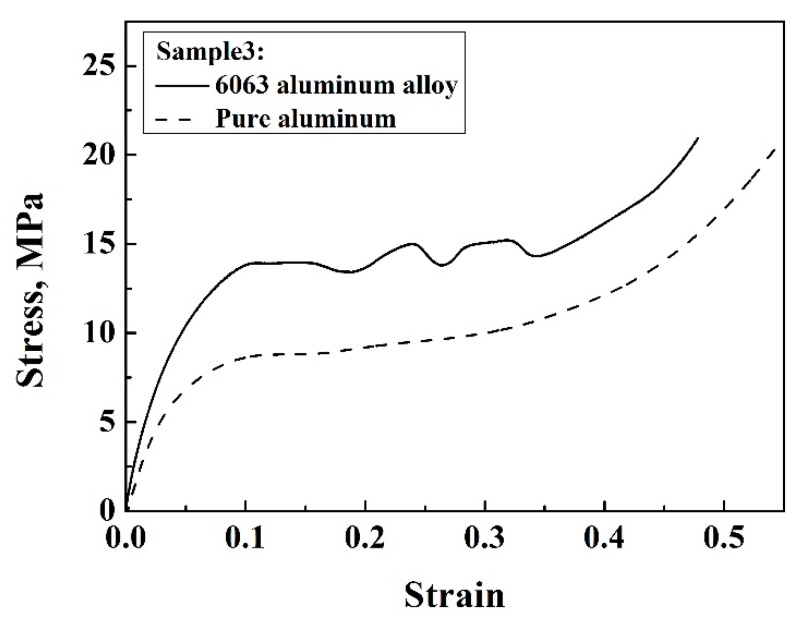
Stress–strain curves of sample 3 with different base materials.

**Figure 11 materials-13-01008-f011:**
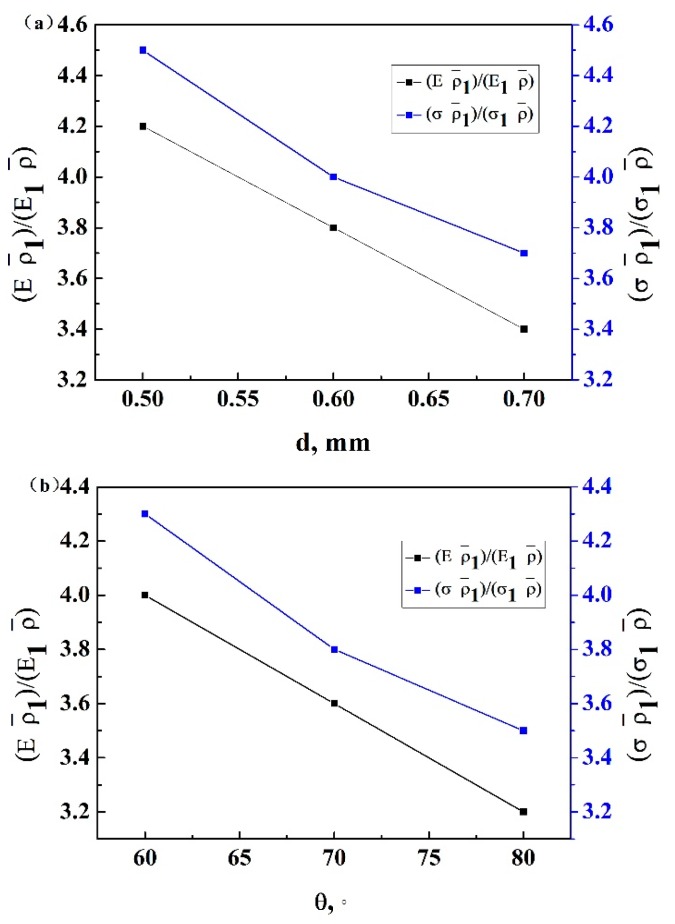
The enhancement of the specific stiffness and strength of the structures with different (**a**) connecting strut diameters and (**b**) *θ* angles.

**Figure 12 materials-13-01008-f012:**
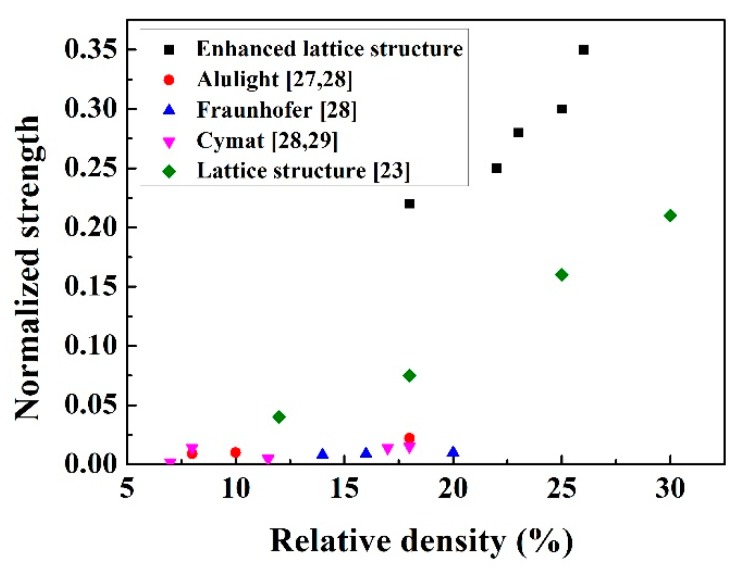
Normalized compressive strength of the enhanced lattice structures and other cellular materials [27,28,29].

**Table 1 materials-13-01008-t001:** Structure parameters of the 3D enhanced auxetic lattice structure.

Sample	H (mm)	L (mm)	θ (°)	D (mm)	d (mm)	Relative Density	
						$\mathrm{Theoretical}\;{\bar{\rho }}_{t}$	$\mathrm{Experimental}\;\;\;{\bar{\rho }}_{e}$
1	7.0	3.5	70	1.4	0.7	0.27	0.25
2					0.6	0.25	0.23
3					0.5	0.23	0.22
4	7.0	3.5	60	1.4	0.5	0.28	0.26
5			70			0.25	0.22
6			80			0.19	0.18

**Table 2 materials-13-01008-t002:** Structure parameters of the 3D re-entrant lattice structure.

Sample	H (mm)	L (mm)	θ (°)	D (mm)	Relative Density	
					$\mathrm{Theoretical}\;{\bar{\rho }}_{t}$	$\mathrm{Experimental}\;\;{\bar{\rho }}_{e}$
7	7.0	3.5	60	1.4	0.24	0.24
8			70		0.20	0.20
9			80		0.16	0.15

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
