# Peer review of "An Enhanced Three-Dimensional Auxetic Lattice Structure with Improved Property"

_materials, 2020, doi:10.3390/ma13041008_

Round 1
Reviewer 1 Report
In this work authors designed and fabricated a series of lattice structures with different morphologies via 3D printing combined with molten metal infiltration technique. They then conducted parametric studies to explore the enhanced auxetic lattice structure to achieve superior mechanical behaviors compared to the 3D re-entrant lattice structure. 3D printing and auxetic structures are among the most important technological problem, and such that this study is timely and the obtained results can be potentially useful and attractive. The study is also well-conducted and the obtained results are worthy for publication. I can thus recommend the publication of this manuscript in its present form.Author Response
Thank you very much for the comments on our manuscript.
Reviewer 2 Report
The manuscript presents interesting study about mechanical properties of auxetic materials. Presented results are interesting but I have several remarks to the manuscript:
1) The axes labels are very small and it make them extremly hard to be read.
2) The microstructures of tested materials should be provided. In the manuscript are only schematic drawings but not presentation of the actual structures.
3) The behaviour of Al alloy should be better descibed (especially the peaks in plastic region).
4) English quality should be improved. Not only it is disturbing but sometimes it even makes hard to understant the text (e.g. sample 3 with same relative density).
Reviewer 3 Report
In this paper, the authors explore the mechanical properties of 3-D auxetic lattice structures under compressive loading. The main focus of the paper is the study of the property effects of changing the design parameters using the described AM-infiltration method. This is certainly a topic of interest, the experiments are well-designed and seem to be performed well (even though incompletely analysed, as discussed below) and the paper is generally well written (though the authors did not follow the journal template in preparing the paper). However, this paper is very similar to a wide variety of other works, several that I am familiar with that are written by the same authors. While a good approach to studying the problem which clearly provides useful data, it is not completely clear what is new and novel in this work and why it is a large and valuable enough contribution for an archival journal article. Simply using infiltration on a well-known lattice problem without more analysis and theoretical modeling/exploration isn't necessarily a novel contribution - the authors should explain this in more detail.
Since this group of authors has a number of previous studies in this area, it is important that they be more clear what they have previously accomplished and what is novel about this work. In addition, the research team for Reference 16 (no affiliation with the reviewer) has a lot more contributions in this area that need to be acknowledged in the literature review.
Detailed comments:
1. The title is too generic, too similar to other works in this area, and should be improved
2. The authors should provide some kind of roadmap for the article at the end of the introduction section. On first reading, the organization of the paper is confusing. It makes sense once read, but should be pointed out to the reader at the beginning.
3. This work seems to try to approach the problem from design, manufacturing, and materials perspectives and I do not think any of them are well-enough represented to carry the paper on their own. Therefore, the authors should decide which one they wish to focus on and restructure the paper toward it. If design, please provide more details about potential design/optimization functions and constraints, if manufacturing, provide more details about the manufacturing and an evaluation of the approperiatness of using the selected AM processes and materials, etc.
4. While I understand that the authors do not intend to present any simulation results in the paper, it would be helpful to include a simple FEA of the lattice structures used. Just detailed enough to generate some expected outcomes that can be discussed later. Based on the experimental results, it is probably acceptable to use a standard linear material model for this, even though the outcome should be nonlinear as shown in the experiments.
5. I would like to see a lot more detail about the manufacturing process used to generate the lattices (regardless of the paper's perspective) and about any problems or events encountered. Infiltration is not a common technique for many applications and have the potential to have very good or bad outcomes depending on the quality of the process.
6. It would be very good to show some of the samples during and after loading so the reader can derive some information about how the failure may have occurred, etc.
7. Do the authors have any reason to believe that residual stresses during the AM processes played a role in the results? If not, why not?
8. Was any fracture or cracking behavior observed in the structure during failure? Was the failure due to bucking or plastic deformation in the members? You certainly have a highly non-linear failure model (e.g., Figures 5, 6, 8, 9) and need to explore why that is.
___________________________________________________________
With these points in mind, I recommend that the editor return the paper to the authors for a major revision to address these points. I hope these comments are helpful and I look forward to seeing the next iteration of this work.
Round 2
Reviewer 2 Report
The manuscript was impruved according to suggestions of both reviewers.
Reviewer 3 Report
The authors have adequately responded to my concerns with the study and manuscript and I have no further comments on the paper. Recommend acceptance in current form.